# Insights into the genetic diversity, recombination, and systemic infections with evidence of intracellular maturation of hepadnavirus in cats

Chutchai Piewbang[1,2], Sabrina Wahyu Wardhani[2,3], Surangkanang Chaiyasak[4], Jakarwan Yostawonkul[3,5], Poowadon Chai-in[5], Suwimon Boonrungsiman[5], Tanit Kasantikul[6], Somporn Techangamsuwan[1,2]*

1 Department of Pathology, Faculty of Veterinary Science, Chulalongkorn University, Bangkok, Thailand, 2 Animal Virome and Diagnostic Development Research Group, Faculty of Veterinary Science, Chulalongkorn University, Bangkok, Thailand, 3 The International Graduate Course of Veterinary Science and Technology (VST), Faculty of Veterinary Science, Chulalongkorn University, Bangkok, Thailand, 4 Office of Academic Affairs, Faculty of Veterinary Sciences, Mahasarakham University, Mahasarakham, Thailand, 5 National Nanotechnology Center (NANOTEC), National Science and Technology Development Agency (NSTDA), Pathumthani, Thailand, 6 Department of Preclinic and Applied Science, Faculty of Veterinary Science, Mahidol University, Nakhon Pathom, Thailand

* somporn62@hotmail.com

**Data Availability Statement:** DCH sequences have been deposited in NCBI GenBank under accession

## Abstract

Hepatitis B virus (HBV) is a human pathogen of global concern, while a high diversity of viruses related to HBV have been discovered in other animals during the last decade. Recently, the novel mammalian hepadnavirus, tentatively named domestic cat hepadnavirus (DCH), was detected in an immunocompromised cat. Herein, a collection of 209 cat sera and 15 hepato-diseased cats were screened for DCH using PCR, resulting in 12.4% and 20% positivity in the tested sera and necropsied cats, respectively. Among the DCH-positive sera, a significantly high level of co-detection with retroviral infection was found, with the highest proportion being co-detection with feline immunodeficiency virus (FIV). Full-length genome characterization of DCH revealed the genetic diversity between the nine Thai DCH sequences obtained, and that they phylogenetically formed three distinct monophyletic clades. A putative DCH recombinant strain was found, suggesting a possible role of recombination in DCH evolution. Additionally, quantitative PCR was used to determine the viral copy number in various organs of the DCH-moribund cats, while the pathological findings were compared to the viral localization in hepatocytes, adjacent to areas of hepatic fibrosis, by immunohistochemical (IHC) and western blot analysis. In addition to the liver, positive-DCH immunoreactivity was found in various other organs, including kidneys, lung, heart, intestine, brain, and lymph nodes, providing evidence of systemic infection. Ultrastructure of infected cells revealed electron-dense particles in the nucleus and cytoplasm of hepatocytes, bronchial epithelial cells, and fibroblasts. We propose the intracellular development mechanism of this virus. Although the definitive roles of pathogenicity of DCH remains undetermined, a contributory role of the virus associated with systemic diseases is possible.

numbers MT506039-MT506047. All other relevant data are within the manuscript and its Supporting Information files.

**Funding:** C.P. was supported by the Ratchadapisek Somphot Fund for Postdoctoral Fellowship, Chulalongkorn University. S.W. is financially afforded by scholarship program for ASEAN countries, Chulalongkorn University. This research was funded by The Thailand Research Fund (RSA6180034), Grant for Joint Funding of External Research Project, Ratchadaphisek Somphot Endowment Fund and Veterinary Science Research Fund (RES_61_364_31_037), Chulalongkorn University, and Veterinary Pathogen Bank, Faculty of Veterinary Science, Chulalongkorn University.

**Competing interests:** The authors have declared that no competing interests exist.

## Introduction

For several centuries, blood-transfused recipients had the risk of suffering from post-transfusion hepatitis from an unknown cause, resulting in further investigation of the pathogen associated with this scenario [1]. The subsequent discovery of a hepadnaviral genome, named as human hepatitis B virus (HBV), was an important breakthrough towards enhancing global health concerns by facilitating the establishment of a blood-screening protocol prior to transfusion [2, 3]. The recognition of HBV in the blood of infected humans lead to the further discovery of HBV-like viruses in other animals, not only in mammals but also in avian and fish species [4, 5]. Recently, during transcriptomics studies of an Australian leukemia cat infected with feline immunodeficiency virus (FIV), a novel hepadnavirus, named domestic cat hepadnavirus (DCH), was detected. This merits further study of its natural history, epidemiology, and potential pathogenic roles in domesticated cats [6].

The DCH is a partially double stranded, circular, small DNA virus belonging to the *Orthohepadnavirus* genus, *Hepadnaviridae* family. It has a genome of approximately 3.2 kb of compact DNA, which encodes in four overlapping open reading frames (ORFs) for the polymerase (L), surface (S), core (C), and X proteins, similar to other hepadnaviruses. The three available genome sequences derived from DCH-positive domestic cats in Australia [6], Italy [7], and Malaysia (GenBank accession no. MK902920), show 97.0–98.3% nucleotide similarity to each other. Further identification and genetic characterization of DCH in these and other regions is required for fundamental information regarding the geographical distribution and evolution of this HBV-like virus in cats.

Since the first identification of DCH in a FIV-positive cat in Australia in 2016, surveys of DCH prevalence in cat sera were conducting in Australian and Italian domestic cats, and revealed 6.5% and 10.8% DCH positive cats with active viremia [6, 7]. Interestingly, there is a correlation between DCH-positive cats with suspected clinical symptoms of infectious diseases, particularly in cats with evidence of liver damage [7]. This finding indirectly supported the possible role of the DCH-induced feline liver disease, similar to the nature of the HBV infection in humans that is associated with chronic hepatitis and contributes to hepatocellular carcinoma (HCC) development. This hypothesis was authenticated by a recent international, multicenter, retrospective investigation of DCH in 71 formalin-fixed, paraffin-embedded (FFPE) biopsies from cats with selected liver diseases in specimens obtained from multiple continents, including USA, Australia, New Zealand, and UK [8]. The results revealed a high detection rate of DCH (43%) in chronic hepatitis cats and 28% in HCC cats, while bile duct associated diseases and normal histological livers showed PCR negative tests. Furthermore, DCH localization was observed by *in situ* hybridization in areas of histologically-determined liver inflammation and neoplasia. Together, these support the relatedness between the presence of DCH and feline liver diseases.

However, neither the viral tropism nor the distribution of the DCH in the other organs of infected cats has yet been investigated, and due to the limited studies of DCH biology, and so the degree of tissue tropism, whether there is any correlation between viral replication and liver damage throughout systemic infections, and the genetic diversity of DCH have all yet to be elucidated. The pathobiology of DCH infections in cats needs further determination in order to better understand its DCH infective pattern, and the identification of which organs support DCH, to elucidate viral shedding and transmission. Epidemiological data are also required to facilitate the design of prevention strategies for feline healthcare.

Therefore, we comprehensively investigated the presence of the DCH in randomized cat sera using conventional PCR and described postmortem findings through viral distribution, tissue localization, and viral loads of DCH-positive cats using quantitative (q)PCR and

immunohistochemistry (IHC). Transmission electron microscopy (TEM) was used to confirm the presence of viral particles in the liver and lungs with possible reference to virus maturation and the shedding process. Genetic analysis, based on the full-length genome, revealed that genetic diversity and novel genetic clades of DCH are now circulating in the cat population in Thailand. Possible roles of systemic infections with ultrastructural localization of DCH were also addressed.

## Materials and methods

### Animals and samples

In total, 209 randomized domestic cats, which had visited veterinary hospitals and healthcare services in Thailand during December 2016 to December 2019, were enrolled in this study. Serum samples were collected and kept at -80 $^{\circ}$C for further investigation. Essential signalments, including sex and age, were also recorded for further interpretation. Each cat's age was categorized into one of the five groups of: kittens (0–6 months), junior (7 month–2 y), adult (3–6 y), mature (7–10 y), and senior ($\geq$ 11 y), according to the life stage classification of The American Animal Hospital Association guideline [9].

Fifteen domestic cats, naturally moribund from hepatic-associated diseases, submitted for routine necropsy at Department of Pathology, Faculty of Veterinary Science, Chulalongkorn University during November 2019 to January 2020, also were included. Selected tissue samples, including the brain, heart, intestines, kidneys, liver, lung, lymph node, and spleen, were collected, immersed in 10% (v/v) neutral buffered formalin, and submitted for histopathological processes within a few consecutive days. Sections were stained for Hematoxylin and Eosin (H&E) and Masson-Trichrome. Microscopic evaluations were performed by an American board-certified veterinary pathologist (TK). Fresh tissues, including heart, intestines, kidneys, liver, lung, spleen, and urinary bladder, were additionally collected for further analysis. All procedures were performed in accordance with the guidelines and regulations following the approval of the Chulalongkorn University Animal Care and Use Committee (No. 1931036). The cat's owners gave their written consent for sample collection and data for this study.

### Viral nucleic acid extraction and selected molecular virology tests

Serum samples from 209 living cats and homogenized fresh tissues from the 15 necropsied cats were individually extracted for total viral nucleic acids using a commercial extraction kit (Geneaid, Taipei, Taiwan). The quality and quantity of the extracted total nucleic acids were determined using a spectrophotometer to measure the $A_{260}/A_{280}$ ratio (NanoDrop, Thermo Scientific™, USA), and then kept at -80 $^{\circ}$C. Known feline retrovirus DNA, which are feline leukemia virus (FeLV) and FIV, were initially screened using PCR assays as previously described [10]. Furthermore, pan-family virologic PCR panels for detections of bocaviruses [11], caliciviruses [12], coronaviruses [13], herpesviruses [14], paramyxoviruses [15, 16], and parvoviruses [17] were tested on fresh tissue samples from necropsied cats (as mentioned above).

### Hepadnaviral screening in sera and viral load quantification in fresh tissues

The presence of hepadnavirus in the serum and fresh tissue samples was screened for by conventional PCR using primers specific for pan-hepadnaviral family as previously described [5]. Subsequently, the pan-hepadnaviral PCR positive samples were subjected to another conventional PCR for specific DCH amplification [6]. PCR products were subsequently visualized by running on the QIAxcel capillary electrophoresis (QCES) platform using a QIAxcel DNA Screening Kit (Qiagen, Hilden, Germany) with the previously reported setting and analysis

[18]. The positive amplicons, either from the pan-hepadnaviral PCR or DCH-specific PCR, were then purified using NucleoSpin® Extract II kit (Macherey-Nagel, Düren, Germany) and further submitted for commercial bi-directional Sanger sequencing (Macrogen Inc., Incheon, South Korea) to confirm the specificity of the PCR results.

To quantify the DCH loads, fresh tissues of moribund cats that were previously found to be positive by PCR were further assayed using KAPA SYBR® Fast qPCR Master Mix (2X) Universal kit (KAPABIOSYSTEMS, Sigma-Aldrich, South Africa) with the designed DCH-specific primers [7]. Briefly, 5 µL of extracted DNA (from 1 g of tissue) were added into the 25-µL reaction master mix, containing 200 nM of each primer and 2.5 mM of $MgCl_2$. The qPCR reaction was performed in a Rotor-Gene Q real-time PCR cycler (Qiagen GmbH, Hilden, Germany) with thermal cycling as 40 cycles of 95˚C for 3 s and 60˚C for 20 s acquiring the fluorescence A green. The software reported the cycling A green and melt A green compared with the synthesized positive control: a standard plasmid (TOPO™ TA Cloning™ Kit for Sequencing with One Shot™ TOP10 Chemically Competent *E. Coli*) (Invitrogen, MA, USA) containing the designed 700-bp fragment of the L gene of the three available DCH reference sequences (GenBank accession nos. MH307930, MK117078, and MK902920). The DCH copy number in each sample was calculated on the basis of standard curve calculation generated from 10-fold dilutions of the plasmid.

## Full-length genome characterization and phylogenetic analysis of detected DCH

To characterize the detected DCH strains in the Thai cat population, PCR products derived from six positive serum samples and three moribund cats were submitted for full-length genome sequencing using multiple gel-based PCR amplifications and primer sets as described previously [6], plus the new primers designed from alignment of the available DCH sequences deposited in GenBank (S1 Table and S1 File). The PCR products were submitted for standard Sanger sequencing after visualization in 2% (w/v) agarose gel electrophoresis.

All derived sequences were aligned to the published DCH strains, then assembled and used to construct the full-length genome using BioEdit version 7.0.5.3. Moreover, to assure the correctness of the initial alignment, the derived DCH Thai strains were further aligned with the published DCH strains using the MAFFT sequence alignment program version 7. The sequences of the nine obtained Thai DCH strains were then compared with the three available DCH strains using a pairwise nucleotide identity matrix software embedded in BioEdit. Phylogenetic analysis of full-length DCH genomes and related hepadnaviruses reported in various species were constructed using the maximum likelihood (ML) algorithm with the general-time-reversible (GTR) model, derived from find-best-fit model analysis based on Bayesian Information Criterion (BIC) of nucleotide substitution in MEGA 7 software package. Phylogenetic trees of full-length genome sequence of Thai DCH strains were constructed with 1,000 bootstrapping replicates using MEGA 7.

## Recombination in Thai DCH strains

To explore the role of genetic recombination in the evolution of DCH diversity, a dataset of alignments of all published full-length DCH genomes and those obtained in this study were examined using the various recombination detection methods. The dataset was initially screened for recombination events using the Genetic Algorithm for Recombination Detection (GARD) software [19] as a pre-processing step for selection of interference. The positive recombination sequences were then further examined using the incorporated recombination detection program 4 (RDP4) package v. Beta 4.94 software [20], which included a range of

recombination detection methods: i.e. Bootscan, Chimera, GeneConv, MaxChi, RDP, SiScan, and 3 Seq. In all cases, a cut-off $p$-value of 0.01 and a default setting of Bonferroni correction were used. Due to the high sensitivity of these methods and the abundant signals detected during testing, only sequences that showed a positive recombination event in at least four methods were considered as potential recombination strains. The potential recombination strains identified in the RDP with its major and minor putative parents used as a query sequence, were then subjected to further confirmation of the recombination breakpoint using the similarity plot and bootscaning analysis embedded in the SimPlot software package v. 3.5.1 [21]. The similarity plot and bootscaning analysis were performed with a window and step sizes of 200 bp and 20 bp, respectively. The recombination breakpoints were detected and evaluated by the Kimura-2 parameter (K2P) and the GapStrip models for bootscan analysis similarity plot, respectively [12].

## Cellular and tissue localizations of DCH using IHC

To display the tissue tropism of the DCH, the three necropsied cats that were positive in both the conventional and qPCRs specific for DCH were then subjected to this IHC-based study. The available FFPE tissues derived from these cats, including the brain, lung, liver, lymph node, spleen, heart, kidneys, and intestine, were used for DCH-tissue identification using a horseradish peroxidase (HRP) polymer-conjugated method. The 4-μm-thick FFPE sections were deparaffinized, rehydrated, and subsequently pre-treated by autoclaving at 121 $^{\circ}$C in Tris-EDTA buffer, pH 9.0 for 30 min. The slides were then treated with 3% (w/w) hydrogen peroxide ($H_2O_2$) and 5% (w/v) skim milk in 1% PBS to block any endogenous peroxidase activity and non-specific reactions. Treated sections were then incubated with a 1:8000 dilution of rabbit polyclonal anti-human HBV core (C) antigen (HBcAg; B0586, DAKO, Glostrup, Denmark) at 4 $^{\circ}$C overnight in a moist chamber. After triple washes, the sections were incubated with the EnViSion system as per the manufacturer's recommendations, and subsequently immersed in 3,3'-diaminobenzidine (DAB) and counterstained with Mayer's hematoxylin. A known HBV infected-liver biopsy section from human (kindly provided by Institute of Pathology, Department of Medical Services, Ministry of Public Health) served as a positive control. Incubation with normal rabbit IgG antibody NI01 (Sigma-Aldrich, MO, USA) and no-primary antibody stained incubation slides served as negative controls. Additionally, to further support the results of the manual IHC analysis, an automated IHC with different antigen retrieval and endogenous peroxidase blocking methods was performed on these sections, as detailed in S2 File.

## Western blotting of anti-HBcAg antibody with the DCH

The cross-reactivity of anti-HBcAg antibody against DCH protein was elucidated by western blotting. Liver samples of three DCH-positive cats (CP1N-CP3N) were subjected for protein extraction using RIPA Lysis and Extraction buffer (Thermo Scientific™, USA) following manufacturer's suggestions. The extracted proteins were calculated using Pierce™ BCA Protein Assay kit (Thermo Scientific™, USA) and adjusted to 5 μg/μL with the lysis buffer. The western blotting was performed as previously described with some modifications [22]. Briefly, proteins (125 μg/well) were separated by SDS-PAGE using 12% polyacrylamide gels, electrophoretically transferred to a nitrocellulose membrane, and blocked with 5% skim milk in Tris-buffered saline for 2 h. The proteins were then incubated with the anti-HBcAg antibody (1:2,000; HBcAg; B0586, DAKO, Glostrup, Denmark) for overnight at 4 $^{\circ}$C. Bound reactivity was determined by using horseradish peroxidase (HRP)-labeled secondary antibodies (1:10,000; goat anti-rabbit IgG secondary antibody; ab205718, Abcam, Cambridge, UK) for 2 h at room

temperature. The proteins of interest were visualized with ECL western blotting detection reagents (GE Healthcare) and the western blot imaging was performed using a Pop-Bio Vü Imaging System (Pop-Bio Imaging, Cambridge, UK). HBV-infected human liver (no. 158-liver; kindly provided by the Research Unit of Hepatitis and Liver Cancer of *Chulalong-korn* University, Bangkok, Thailand) was served as a positive control while two liver samples derived from DCH PCR-negative cats with and without histological evidences of hepatitis (nos. 045- and 046-liver, respectively) were served as negative controls.

## Ultrastructure of DCH-infected tissues using TEM

In order to ultra-structurally demonstrate the DCH-viral particles in the cat's tissues, the FFPE sections of the liver and lung from a necropsied cat (cat no. 3) that was DCH-positive in both the PCR and IHC analyses, were subjected to TEM analysis. The TEM samples were prepared from histological slides with double heavy metal staining [23, 24] and pop-off techniques with modifications [25, 26]. Briefly, histological slides were deparaffinized in 100% xylene to dissolve all the paraffin wax, washed in absolute ethanol twice, and then rehydrated in a graded decreasing concentration of ethanol [100 to 70% (v/v)] in deionized water ($dH_2O$). Afterwards, the sections were bulk-stained in staining solution containing 2% (v/v) osmium tetroxide (Electron Microscopy Sciences EMS, PA, USA), 1.5% (w/v) potassium ferrocyanide (Sigma Aldrich, MO, USA), and 2 mM $CaCl_2$ in $dH_2O$ for 1 h and then washed in $dH_2O$. Subsequently, the sections were immersed in 0.5% (w/v) thiocarbohydrazide solution (Sigma Aldrich, MO, USA) in $dH_2O$ for 20 min, followed by 2.0% (v/v) $OsO_4$ in $dH_2O$ for 30 min at room temperature. The sections were then immersed overnight in a solution of 0.15 M gadolinium acetate and 0.15 M samarium acetate (Sigma Aldrich, MO, USA), washed twice in $dH_2O$, and incubated in 1% Walton's lead aspartate at 60 $^0$C for 30 min [27, 28] to enhance the contrast. Sections were then dehydrated in a graded [70 to 100% (v/v)] ethanol concentration in $dH_2O$. The sections were infiltrated with Epon 812 (Electron Microscopy Sciences EMS, PA, USA) and then placed upside down on top of molding capsules and polymerized at 70 $^0$C overnight. The sections were then cracked and removed from glass slide via the pop-off technique. The ultrathin sections were cut and collected using an ultra-microtome (EM UC7, Leica Microsystem, PA, USA). The ultrastructure was investigated using TEM (HT7800; Hitachi, Tokyo, Japan) operated at 80 kV.

## Statistical analysis

All categorical variables were analyzed among groups by Fisher's exact test or the $\chi^2$ test with Yates correction. All tests were two-sided, and a *p*-value of $< 0.05$ was considered to indicate statistical significance. The statistical analyses were performed using the GraphPad Prism 8 and Microsoft Excel for Microsoft Office 365 software.

## Results

### Detection of DCH in cat serum and liver

A total of 209 serum samples were randomly enrolled in this investigation. Eighty-seven subjects (41.6%) were male, and 122 (58.4%) were female; the mean age (± SD) was 3.7 ± 2.7 y, and the median age was 3 y (range: 1 month– 13 y). Among 209 sera, 26 (12.4%) sera were positive to both pan-hepadnaviral PCR and DCH-specific PCR methods. In addition, 15 (57.7%) DCH-positive cats were positive for FIV, 3 (11.5%) DCH-positive cats were individually positive to FeLV. while only one (3.8%) DCH-positive serum was positive to both FIV and FeLV DNA PCRs. There were significant differences in the co-detection of feline retroviruses

($p$ = 0.017), with the highest significance for FIV co-detection ($p$ < 0.00001). However, the sex and age of cats were not significantly associated with DCH infections. All DCH-PCR positive samples were further sequenced and confirmed for the presence of a partial fragment of the DCH genome.

For DCH detection in necropsied cats, 15 fresh liver samples derived from 15 moribund cats showing macroscopic lesions in liver were included in this study. Nine moribund cats were male, and six were female, with a mean age of 3.3 ± 1.7 y. Of these 15 cats, three (20%; two males and one female; mean age of 3.1 ± 0.8) were positive with pan-hepadnavirus and DCH-specific PCRs, and confirmed to be DCH by Sanger sequencing.

## Genetic diversity, phylogeny, and genetic recombination of DCH circulating in Thai cats

After reconstruction of the whole DCH genome detected in this study, the coding sequence of the Thai DCH strains was found to be comprised of four main ORFs that encoded for the L, S, C and X proteins and formed an approximately 3.2 kb long circular viral DNA structure. The full-length genome sequences of the six Thai DCH strains derived from cat sera were named as DCH isolates CP23S THA/2016, CP54S THA/2016, CP15H THA/2019, CP79H THA/2019, CP87H THA/2019, and CP99H THA/2019 (GenBank accession nos. MT506039, MT506040, MT506042–MT506045), while the three Thai DCH strains from moribund cats were named CP1N-THA/2019, CP2N-THA/2019, and CP3N-THA/2019 (GenBank accession nos. MT506041, MT506046, and MT506047), respectively. Nucleotide pairwise identity revealed that all the Thai DCH strains showed genetic diversity amongst themselves, and a 96.0–98.5% nucleotide similarity amongst the other three strains from other countries, with the highest similarity to the DCH strain from Malaysia (UPM CHV04; MK902920) and the lowest to the Australian DCH (Sydney 2016; MH307930). Interestingly, the majority of Thai DCH strains were divergent and formed novel phylogenetic lineages, separating the phylogenetic tree of DCH strains into three major clades, tentatively named as genotypic groups A to C in this study (Fig 1). The phylogenetic topology separated the previously published DCH and two of the Thai DCH strains (CP2N THA/2019 and CP3N THA/2019) into group A, while the remainder of the Thai strains additionally clustered into two different lineages, groups B and C.

Initial use of GARD suggested that there were genetic recombination events in the data set of DCH alignments. This prompted us to further elucidate the recombination events using multiple statistically analytical programs embedded in the RDP4 software to identify the possible potential recombination breakpoint. The recombination breakpoints were identified in the viral C protein of DCH strain CP23S THA/2016 (genotype C) that were supported by statistical algorithms including GENECONV, BootScan, Maxchi, Chimaera, and 3Seq with a $p$-value of $4.398 \times 10^{-7}$, $4.915 \times 10^{-5}$, $5.833 \times 10^{-4}$, $1.523 \times 10^{-2}$, and $2.785 \times 10^{-5}$, respectively. The recombinant CP23S THA/2016 strain had a DCH strain CP54S THA/2016 (genotype C) and a DCH strain Sydney2016; MH307930 (genotype A) derived from an Australian cat as its putative major and minor parents (Fig 2A). The DCH CP23S THA/2016 strain was assured as recombinant strain by a positive recombination breakpoint in the similarity plot and bootscaning analyses, suggesting that it had the high nucleotide similarity to the DCH strain CP54S THA/2016 (blue line), but also had nucleotide identity to the DCH strain Sydney2016 (red line) at the middle of viral core protein gene (Fig 2B). The putative recombination breakpoint was identified at the location of 1,980–2,178 nt of the C protein gene. The potential recombination event was also confirmed by phylogenetic tree construction of different genome segments (Fig 2C).

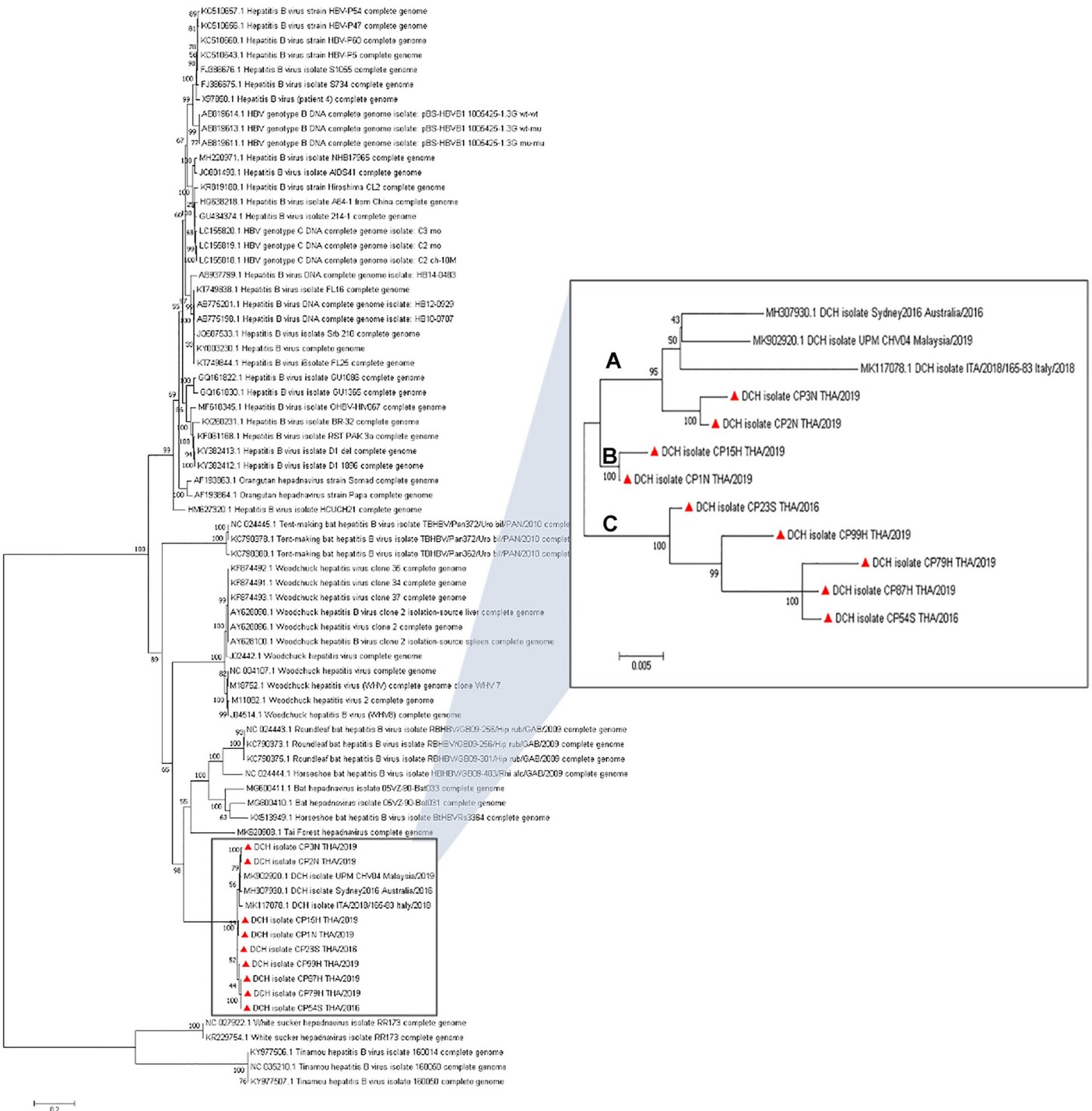

**Fig 1. ML phylogenetic tree showing the genetic relationship of the full-length coding DCH genomes.** The ML tree was constructed using the GTR model and 1,000 bootstrapping. Bootstrap values (%) are shown above nodes when > 50%). Reference sequences used in this analysis are indicated with their respective GenBank accession nos. The DCHs were clustered with mammalian hepadnaviruses, while the avian and fish hepadnavirus sequences served as outgroups. The phylogenetic tree of DCH revealed three different phylogenetic clades (Groups A–C) with high bootstrap support (95–100%). The Thai DCH strains detected in this study are indicated by a red triangle (Inset).

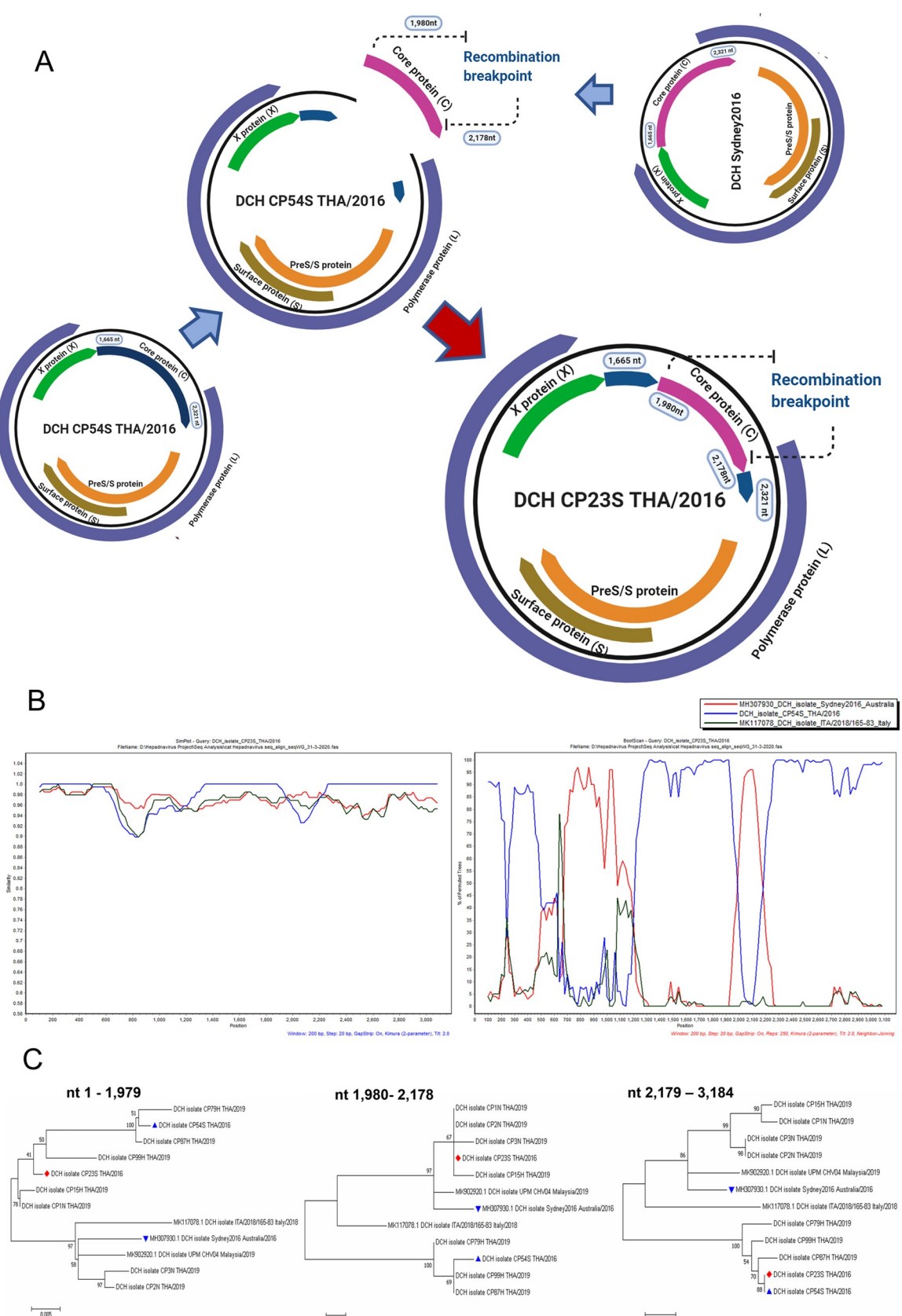

**Fig 2. Schematic diagram of the recombinant DCH CP23S THA/2016 strain.** (**A**) The DCH CP54S THA/2016 isolated in this study and DCH Sydney2016 (MH307903) from Australia served as the putative major and minor parents. (**B**) The potential recombination event was detected in the core protein gene and was supported by similarity and bootscan analysis, which indicated that DCH CP54S THA/2016 (blue line) served as the main template of the whole DCH genome, and then the genome sequence was replaced by the sequence from the DCH Sydney2016 at most of the C gene. The DCH CP23S/2016 isolate served as the query. The y-axis indicated the percentage of nucleotide identity and permutated trees for the similarity plot and boot scanning, respectively, within a 200 bp-wide window with a 20-bp step size between plots. (**C**) The ML phylogenetic trees of the recombinant DCH23S THA/2016 strains (●) and its major (▲) and minor (▼) putative parent strains over three different segments. Bootstrap (1000 replications) values over 50% are shown for each node.

## Viral localization and related pathological changes of DCH-positive moribund cats

Three DCH-positive necropsied cats, designated as cats no. 1 to 3, respectively, were examined. With reference to the collection of fresh tissues, the qPCR targeting for DCH was used to confirm the presence of the virus and to estimate the amount of DCH DNA in various organs. From the qPCR analysis, DCH DNA was identified in the liver, lung, kidneys, urinary bladder, and spleen in all three necropsied cats, while DCH DNA was additionally detected in the intestine of cats no. 1 and 3, and the heart for cats no. 2 and 3. The amount of DCH DNA is shown in Table 1. The presence of DCH DNA in various tissues raised the question as to whether this represented hematogenous spreading or actual viral localization. Accordingly, two independent IHC methods against the hepadnaviral C protein were used to identify the cellular tropism of DCH with respect to the related pathological changes. Overall, within several sections of three necropsied cats and IHC staining methods, the staining patterns were similar among cats.

Postmortem examination of all three necropsied cats revealed varied degrees of systemic inflammations in various organs. Specifically, hepatitis and hepatic fibrosis were illustrated with different severities in all cats. For cats no.1 and 2, hepatocytes were diffusely rounded and dissociated, the central veins were thickened by moderate bands of fibrous connective tissue, and associated mild to moderate aggregates of hemosiderin laden macrophages, lymphocytes, and plasma cells. The liver section of cat no. 3 showed a greater severity, presented a massive surrounding of thick bands of fibrosis bridging between portal structures (Fig 3A). These dense bands of fibrosis contained high numbers of tortuous and slit-like bile ducts that were associated with increased arteriolar profiles, and specifically the centrilobular hepatocytes were variably atrophied and formed haphazard thin cords. Such pathological changes in the liver sections in all cats were governed by the positive Masson-Trichrome special staining (Fig 3A, inset), suggesting evidence of liver fibrosis. The IHC staining patterns were similar among the three cats, revealing a cytoplasmic positive signal of DCH that was strongly detected in multi-focal and diffuse staining in the hepatocytes and bile duct epithelial cells, adjacent to the focal inflammation and fibrosis (Fig 3B).

In the kidneys, glomerular changes in cats no. 2 and 3 were more prominent than in cat no. 1. The walls of glomerular capillaries of cats no. 2 and 3 were globally diffusely thickened by moderate hyalinized material, while the changes in the glomerular capillary walls of cat no. 1

**Table 1. Copy number of DCH in different organs of three necropsied cats.**

| Cat no. | Viral copy number (genome copies/gram of tissue) | | | | | | |
|---|---|---|---|---|---|---|---|
| | Liver | Heart | Lung | Intestine | Kidney | Urinary bladder | Spleen |
| **1** | $2.11 \times 10^4$ | 0 | $0.18 \times 10^1$ | $2.41 \times 10^7$ | $1.39 \times 10^5$ | <1 | $2.63 \times 10^7$ |
| **2** | $1.37 \times 10^4$ | $2.42 \times 10^1$ | $1.82 \times 10^2$ | 0 | $1.54 \times 10^3$ | <1 | <1 |
| **3** | $1.28 \times 10^8$ | $1.59 \times 10^5$ | $9.30 \times 10^7$ | $1.82 \times 10^9$ | $5.85 \times 10^4$ | <1 | $1.49 \times 10^8$ |

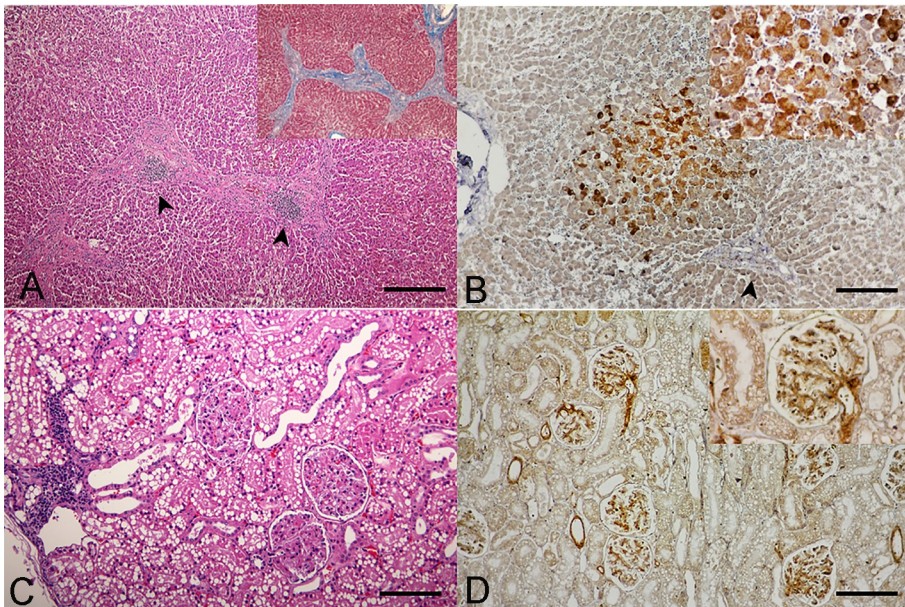

**Fig 3.** Domestic cat hepadnavirus infection of the **(A, B)** liver and **(C, D)** kidney. Representative **(A, C)** H&E and **(B, D)** DHC IHC images from cat no. 3. **(A)** Hepatic fibrosis with sinusoidal lymphocytic infiltration (arrowheads) plus positive staining to Masson-Trichrome special staining (inset). Bar indicates 45 μm. **(B)** DCH-immunoreactivity signals were indicated in cytoplasm of hepatocytes (inset) in the area adjacent to hepatic fibrosis (arrowhead). Bar indicates 170 μm. **(C)**. Membranoproliferative glomerulonephritis with focal interstitial nephritis. Bar indicates 170 μm. **(D)** DCH immunopositivity was observed in the vascular pole, glomerular capillary loops and basement membrane, and mesangial cells (inset). Bars indicate 170 μm.

were mild, and only segmental and multifocal. All cats revealed dense interstitial aggregates of lymphocytes and plasma cells scattered in the cortical region (Fig 3C). The renal tubules were focally replaced and effaced by a thick band of dense fibrous connective tissue containing small aggregates of hemosiderin laden macrophages and interstitial infiltrates of lymphocytes and plasma cells scattered within the renal cortex. The DCH IHC signals were observed in endothelial cells of vascular poles, glomerular capillary loops and basement membrane, and mesangial cells (Fig 3D), where membranoproliferative glomerulonephritis was evident.

For the lung, the changes varied from acute to chronic among the three cats. In cat no. 1, the lung had diffuse interstitial infiltrates of mixed leukocytes, and the alveolar lumens contained variable eosinophilic proteinaceous materials and erythrocytes-laden foamy macrophages. For cats no. 2 and 3, the alveolar interstitium was widely and multifocally expanded by thick bands of fibrosis, the smooth muscle of terminal airways was moderately hyperplastic, and the pulmonary alveolar capillaries were increased in prominence, tortuous, and markedly engorged. The subsequent collapsed alveoli were segmentally lined by plump and tombstoning type II pneumocytes that contained variable numbers of foamy histiocytes. In these regions, many bronchiolar/bronchial airways were obviously collapsed (Fig 4A), the lining epithelium varied from hyperplastic to attenuated and eroded, and there were multifocal follicular aggregates of prominent bronchial-associated lymphoid tissue noted in some areas. The DCH-immunoreactive signals were strong and diffuse staining in the cytoplasm of the bronchial and bronchiole epitheliums, lamina propria connective tissues, vascular endothelial cells, and bronchial gland epitheliums (Fig 4B).

Changes in the heart were revealed as various degrees of myocardial disarray and hypertrophy among the three cats, as evidenced by the haphazard orientation of cardiac myofibers resulting in the formation of branching or radiating "pinwheel" patterns (Fig 4C). Note that

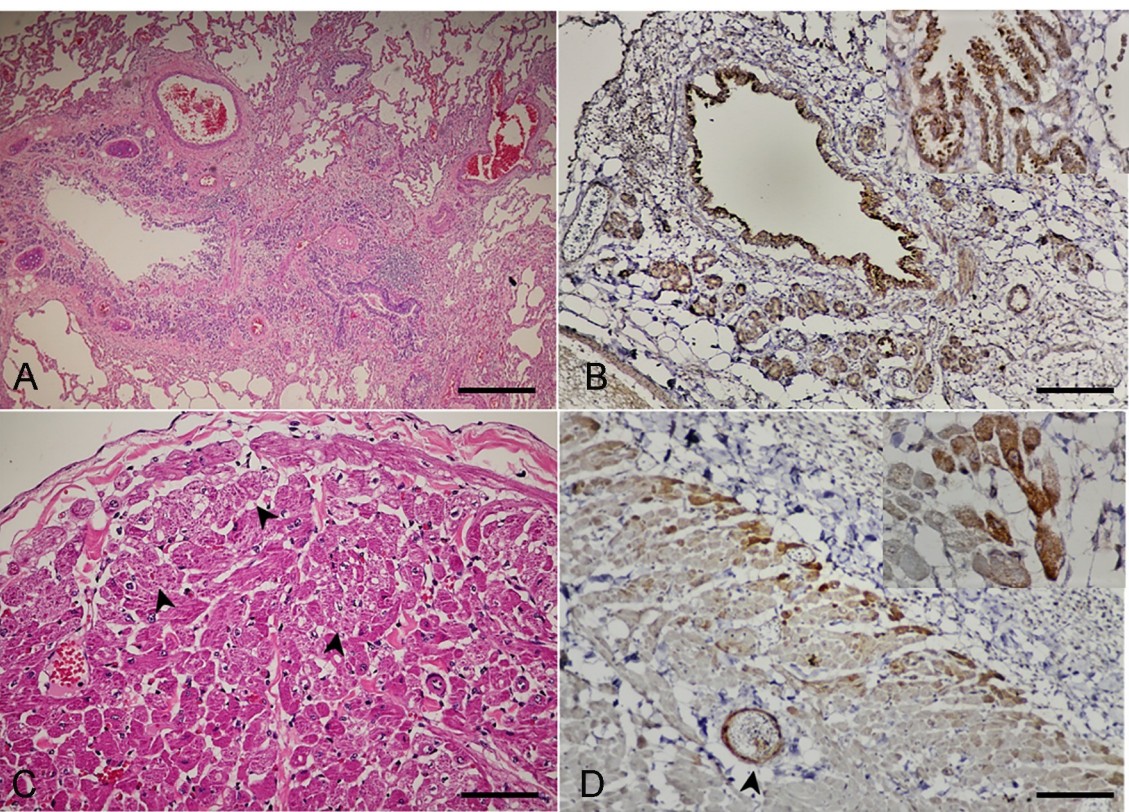

**Fig 4.** Domestic cat hepadnavirus infection in the **(A, B)** lung and **(C, D)** heart. Representative **(A, C)** H&E and **(B, D)** DHC IHC images from cat no. 3. **(A)** Severe broncho-interstitial pneumonia with collapsed bronchial airways. Bar indicates 45 μm. **(B)** DCH-immunoreactivity signals were clearly detected in the cytoplasm of bronchial luminal (inset) and glandular epitheliums and endothelial cells of blood vessels. Bar indicates 45 μm. **(C)** Multifocal cardiac necrosis revealed by the pinwheel pattern (arrowheads) and focal cardiac fibrosis. **(D)** DCH immunopositive-signals were detected in the cytoplasm of myocardium (inset) adjacent to the pericarditis. The endothelial cells were also immunopositive (arrowhead). Bars indicate 170 μm.

for cat no. 3, the epicardium was diffusely thickened by thick bands of fibrosis and granulation tissue containing dense, dissecting, interstitial, and perivascular infiltrates of predominant neutrophils, and a few foamy histiocytes, lymphocytes, and plasma cells dissecting the plump spindloid fibroblasts. The DCH IHC signals were also detected in the myocardium, where evidence of myocardial necrosis was noticed in cat no. 3 (Fig 4D).

Within the sections of intestines and lymph nodes, the histology was similar among the three cats, with moderate lymphoplasmacytic enteritis and histiocytic lymphadenitis. The DHC immunoreactive signals were clearly evident in the cytoplasm of the cryptal epitheliums and in the endothelial cells within the lamina propria in intestinal villi (Fig 5A), as well as diffuse, generalized, intracytoplasmic staining in the histiocytes residing in the lymph nodes (Fig 5B). Although the histology of the cerebellum of necropsied cats revealed unremarkable lesions, the DCH IHC signals were evident in neurons and glia cells (Fig 5C) with moderate and diffuse staining in the endothelial cells of small blood vessels and in choroid plexus epitheliums (Fig 5D). No DCH IHC signal was detected in the negative controls (S1 Fig).

## Cross-reactivity of anti-HBcAg antibody with DCH

To ensure the results of DCH-IHC using anti-HBcAg antibody, the cross-reactivity of the antibody to the DCH was tested using western blotting. In this test, we compared the reactivities

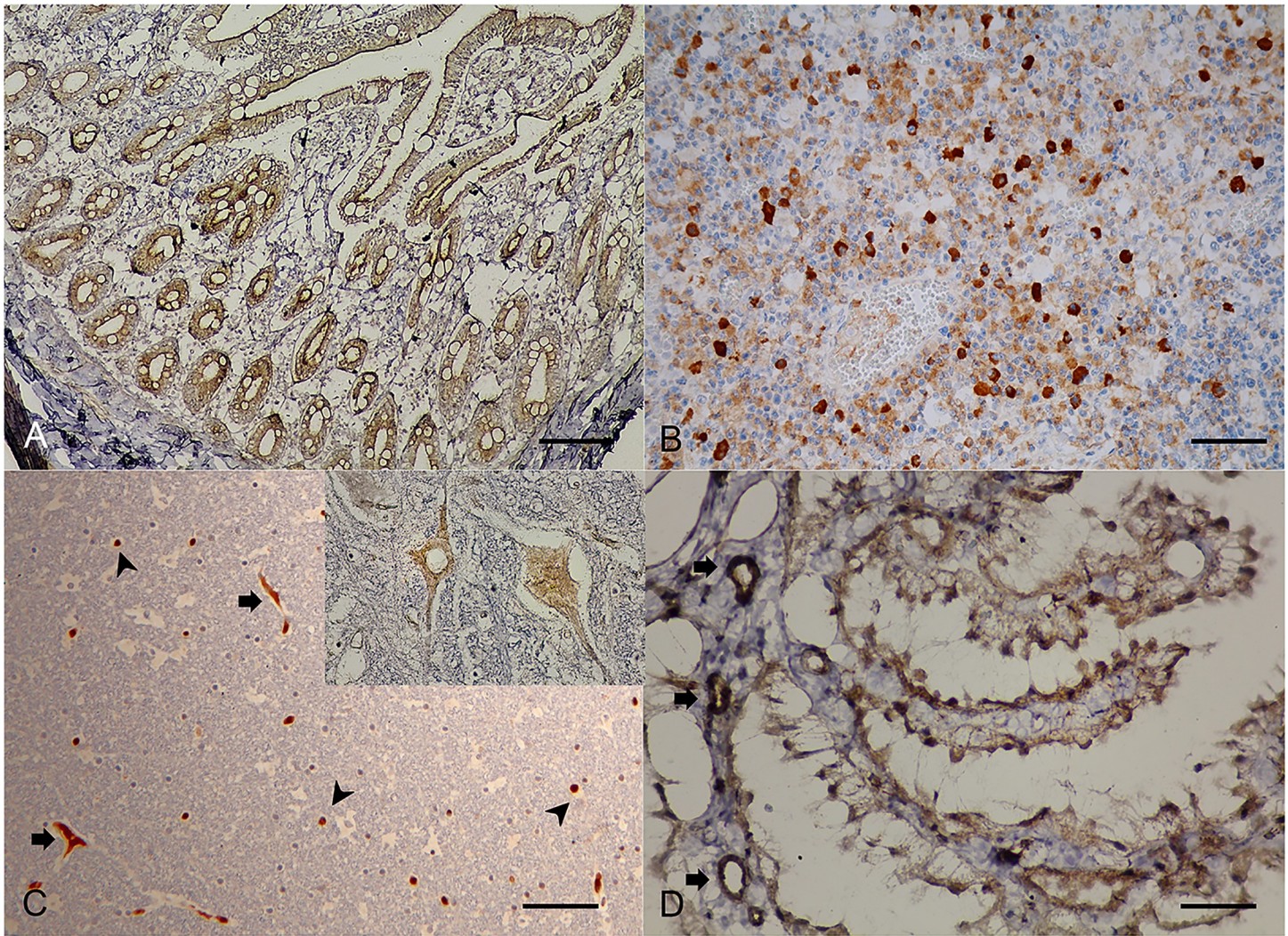

**Fig 5.** Domestic cat hepadnavirus infection. Representative IHC images from (**A, B**) cat no. 2 and (**C, D**) cat no. 1. (**A**) Intestine. Immunopositive signals diffusely labelled in the cytoplasm of cryptal epithelial cells. (**B**) Lymph node. Histiocytes circulating in lymph node were labelled with DHC immunoreactivity. Bars indicate 45 μm and 850 μm, respectively. (**C**) Cerebellum. Immunoreactive signals were labelled in the cytoplasm of neurons (inset) and vascular endothelium (arrows), and in the nucleus and cytoplasm of glia cells (arrowheads). (**D**) Immunopositive signals were also detected at the choroid plexus and relating endotheliums. Bars indicate 45 μm and 850 μm, respectively.

of the antibody towards liver tissue extracts derived from DCH PCR-positive cats (nos. CP1N-CP3N-liver) while the HBV-infected human liver extract (no. 158-liver), and DCH PCR-negative liver sample extracts (no.045-liver; hepatitis cat and no. 046-liver; non-hepatitis cat) were served as positive and negative controls, respectively. As expected, the HBV-positive sample revealed positive reactivity to the about 21 kDa protein band that the molecular weight related to HBV core protein [22, 29]. The immunoreactivity was observed in the DCH-positive samples as in the same target to the HBV core protein at the various intensity among the samples. The positive antigenic protein was absent in all DCH-negative samples (Fig 6).

## Ultrastructure of DCH in tissues

To further verify the results of DCH detection in tissues using qPCR and IHC, TEM was performed on selected lung and liver FFPE samples derived from necropsied cat no. 3, confirming

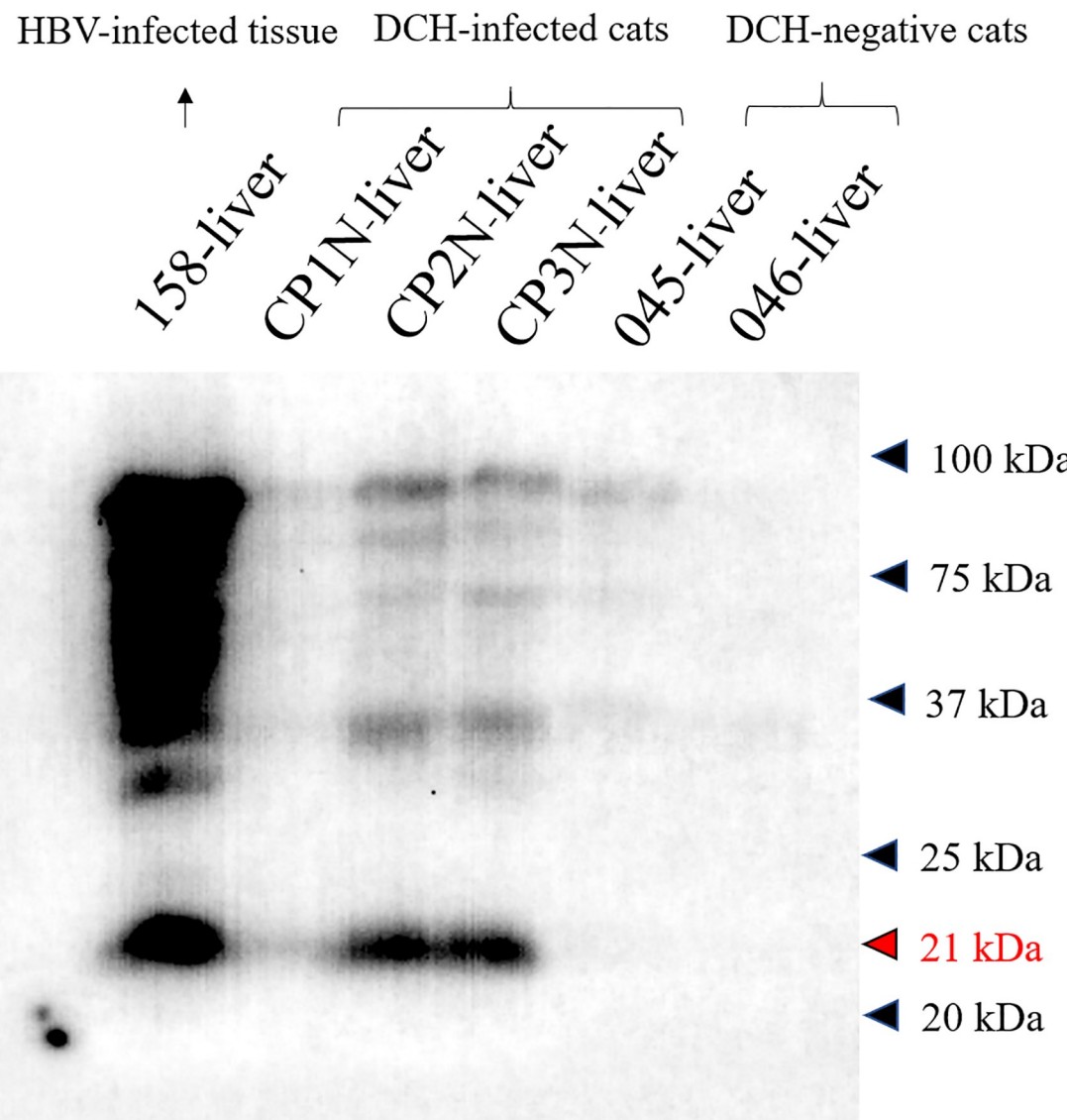

**Fig 6. Cross-reactivity of anti-HBcAg with the DCH.** Western blot analysis revealed positive reactivity of 21 kDa protein of DCH-positive liver samples (CP1N-CP3N-liver) that was similar to result of the HBV-positive sample (158-liver). No reactivity was observed in DCH-negative samples (nos. 045- and 046-liver).

the presence of DCH particles. In the liver section, viral surface antigen spheres, estimated at 23–25 nm in diameter, were identified in the nucleus (Fig 7A) and/or cytoplasm (Fig 7B) of hepatocytes and some biliary ductal cells. Notably, the viral particles were occasionally observed in the nuclear membrane and they were protruding into the endoplasmic reticulum (ER) (Fig 7B), resembling a budding process. The ER membrane appeared to proceed into its lumen to form an invagination, attached with the particles. The invagination continued and enveloped the particles within the wall of ER, envisaged as a particle budding process.

Interestingly, larger viral particles with a double membrane layer, similar to a Dane particle of the HBV, and estimated to be 42 nm in diameter, were seen in the vicinity of the ER (Fig 7B), where its cisternae were dilated and contained viral particles in some areas. Complete Dane-like particles were seen in the cisternae that contained intracisternal tubular structures,

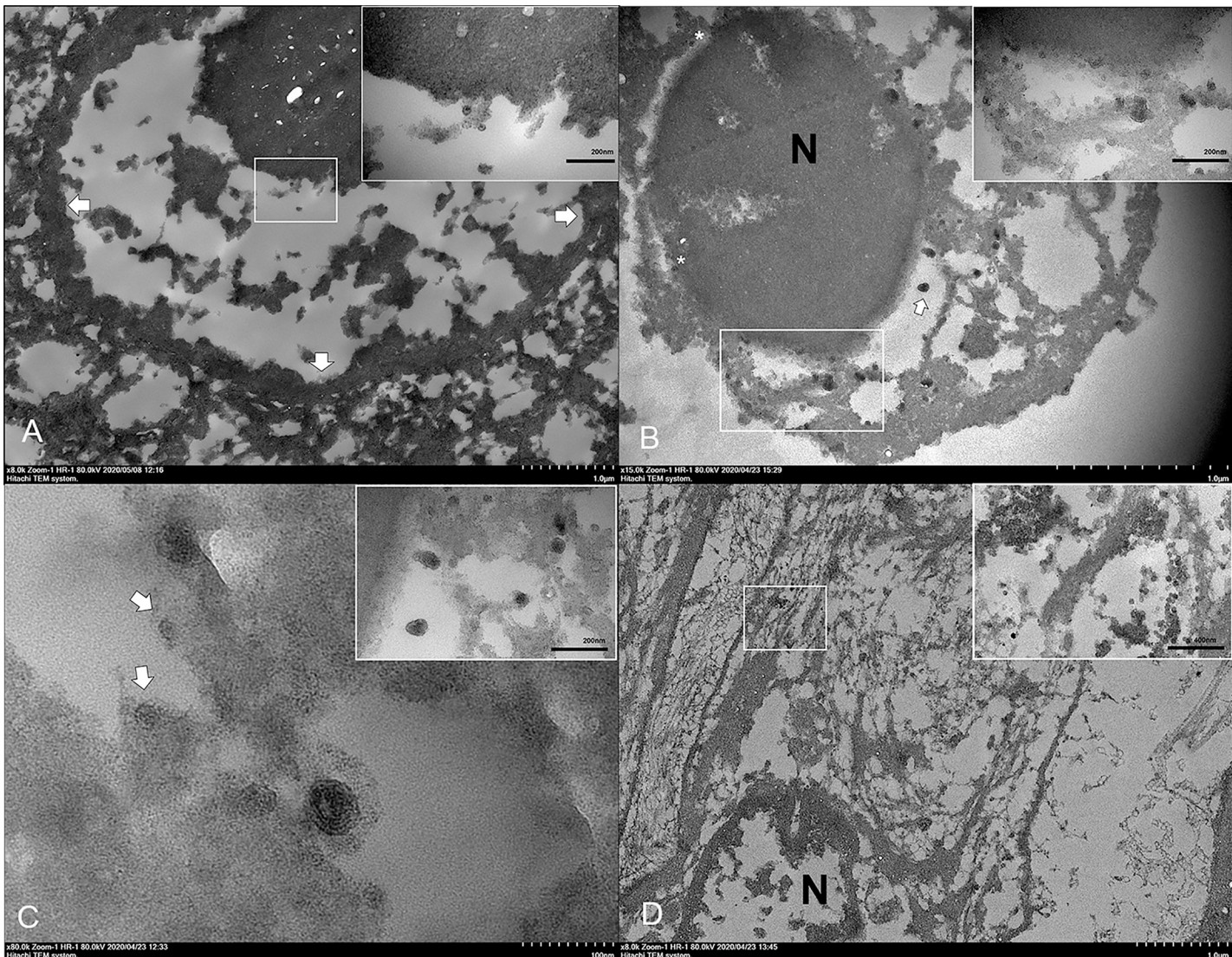

**Fig 7. Ultrastructure of DCH and intracellular development of DCH particles in the liver.** Representative TEM images from the liver of cat no. 3. (**A**) The DCH antigen spheres (inset) were detected in the nucleus of hepatocytes. Arrows indicate the nuclear membrane. (**B**) Numerous DCH particles were found in the nucleus (N) with a single viral sphere inside the nuclear membrane (asterisks). A complete Dane-like particle (arrow) was observed in the cisternae of the ER and numerous incomplete Dane-like particles and viral particles were attached to the ER membrane (inset). (**C**) Invagination of the ER membrane with a complete Dane-like particle. Incomplete Dane-like particles stalked with the ER membrane and attached viral particles with the ER membrane were observed (arrows). The Dane-like particles were frequently observed as free-floating and attaching to the membrane (inset). (**D**) Numerous clusters of DCH core particles (inset) in the cytoplasm of fibrous cells residing in the area of fibrosis. Bars indicate as described in figures.

and presented as a tadpole-like appearance connecting the Dane-like particles with the tubular structures, while some incomplete Dane-like particles were noted to be attached at the ER membrane (Fig 7C). Furthermore, complete Dane-like particles were seen at the outer plasma membrane of hepatocytes and some particles were sheathed by the plasma membrane. Apart from the hepatocytes, the viral particles were frequently observed in the cytoplasm of elongated cells that were surrounded by collagen fiber, indicating these cells were fibroblasts (Fig 7D).

For the ultrastructure of lung section, the viral spheres and Dane-like particles were frequently observed. However, free viral particles were seen in the cytoplasm of bronchial epithelium (S2 Fig) and in the nucleus of bronchial gland (S3 Fig). Of note, numerous free- and

clustered-viral particles were observed in the cytoplasm of fibroblastic cells. Neither viral core particles nor Dane-like particles were observed in other pulmonary cells.

## Discussion

Since the first identification of DCH in a leukemia cat in 2018, further surveys of DCH in Australian and Italian cat sera revealed active DCH viremia in cats showing evidence of feline retroviral infection with increased hepatic enzyme values representing liver damage [6, 7]. These findings provided an indirect mirror on what has been seen in HBV infections in humans and hepadnaviral infections in various animals [4]. A recent retrospective study of DCH in feline FFPE-liver sections using PCR and *in situ* hybridization revealed hepatotropism of the virus and associations with hepatitis and HCC [8], which have been considered to be idiopathic in this species over time [30]. However, the role of DCH pathogenicity in active cases through the cellular distribution of DCH in other organs has not been elucidated. Furthermore, information regarding the genetic diversity and evolution of DCH are clearly limited, due to the fact that the DCH is a recently discovered virus, and so further investigations are needed.

In this study, we investigated the presence of DCH in randomized cat sera and found it has likely been circulating in the Thai cat population since at least 2016 (no samples prior to 2016 were tested), similar to the timeframe of the first identification of DCH [6]. Interestingly, we found a significant prevalence in the cohort of DCH-positive cats with retrovirus co-detections, with the highest significance being with FIV coinfections. These results were in accordance with previous studies that indicated most DCH-positive cats were co-detected with feline retrovirus [6, 7]. Because feline retroviruses cause impairment to the immune system [31, 32], the detection of DCH in cats simply because they are immunocompromised is possible, and is what has been recognized for HBV infection, where HBV is more prevalent in immunocompromised humans [33, 34]. Further investigation of the association of DCH with feline retroviruses is needed to elucidate this observation.

Little is known about the genetic diversity of DCH, which prompted us to study the full-length genome of randomized nine of the detected Thai DCH strains. These sequences revealed a genetic diversity among the Thai DCH strains and amongst the three previously published DCH strains from other countries, forming three separate monophyletic groups, tentatively named in this study as groups A to C. The genetic diversity of DCH, which may result from geographically distinct mutations or recombination among detected DCH strains, has also been displayed in the HBV genotype and subgenotype characteristics [35, 36]. Further investigation into the epidemiology of DCH in other regions by genetic characterization will fill this gap.

Studies on hepadnaviral evolution have indicated the role of genetic recombination in the evolution of these viruses [37–39]. Therefore, we examined potential recombination in the Thai DCH strains and found one likely homologous DCH recombinant (strain CP23S THA/2016), suggesting a possible role of recombination in DCH evolution. The finding of the genetic recombination event at the C gene of DCH CP23S THA/2016 mirrors the favored position for genetic recombination in other hepadnaviruses, which are frequently observed in HBV genotypes in humans and non-human apes [37, 40]. Mixed infections are an influencing factor for genetic recombination [41, 42]. In the present study, the DCH strains (CP2N- and CP3N- THA/2019) were most closely related to DCH Sydney2016, indicating the likely origin (strain) from which the 1,980–2,178 nt region in the C gene of the recombinant DCH CP23S THA/2016 circulating in Thailand came from. Therefore, we speculated that recombination during a mixed infection was the most likely origin of this recombinant DCH strain. However, neither the mechanistic factor that influenced the recombination event nor the effects of this mutation on DCH are known.

A recent study demonstrated the association of DCH in feline hepatitis and HCC, which mirrors that of HBV [8]. However, the pathogenesis and distribution of DCH throughout different tissues was unknown. Accordingly, we described the systemic infection of DCH and associated lesions in various organs of three Thai cats. Overall, the viral localization and histological features of this DCH resembled HBV infection in humans and other hepadnavirus infections in other animals [43–47]. The most consistent lesion found in DCH-positive cats by IHC was chronic active hepatitis and hepatic fibrosis, with the cytoplasmic expression of the DCH-viral C protein in hepatocytes.

Due to the fact that different staining patterns of the HBcAg IHC may reflect differences in the histological activity and viral replication [48–50], the IHC staining pattern was confirmed by a second automated IHC. Together these results suggested that these necropsied cats may be at a chronic stage of the infection, supported by the histological feature of hepatic fibrosis and low DCH viral copy number in the liver. The western blot analysis revealed the presence of viral protein, indicating the expression of the virus in the liver samples of infected cats. Furthermore, the western blot analysis indicated the cross reactivity and specificity of anti-HBcAg antibody to the DCH antigen, supporting the results of IHC. Since we used the polyclonal antibody against HBV C protein to detect the immunoreactivity with the DCH, a clear state for relationship of staining pattern with the stage of infection may be subjected for specificity of this IHC and need further elucidation by using specific monoclonal antibody against DCH.

Apart from liver, we described, by qPCR and the expression of HBcAg IHC, for first time, the DCH localization in various organs and cell types outside of the liver, supporting the spectrum of extra-hepatic manifestations of the nature of hepadnaviral infections [43, 46, 51]. The different DCH viral loads among the investigated cats may support the degree of severity, as presented in HBV-infected human patients [52]. Although this resemblance between HBV and DCH is fascinating, whether a correlation also exists between DCH replication and liver damage still requires to be established. For instance, the varying DCH viral loads in different organs cannot be concluded in this study, and a further structured, larger observational study is required.

Within the glomerulonephritis, we revealed DCH infection in the renal and vascular endothelial cells, supporting that viral localization may be associated with lesions. Like HBV, which makes kidneys its potential reservoir for its replication [53, 54], the possible role of DCH-associated glomerulo-nephropathy requires consideration. Recent observations in HBV-infected humans have speculated on the possibility of HBV replication in nervous systems [44, 55]. Thus, the positive DCH immunoreactivity in the choroid plexus and endothelial cells may support the ability of DCH to cross the blood brain barrier and the possibility that DCH can replicate in these tissues, as supported by the observed DCH immunoreactivity localized in neuron and neuroglia cells.

Similar to the findings in this study, the extrahepatic biology of other hepadnaviruses in animals, including HBV in patients with chronic hepatitis [56] has revealed that various cell types, including endothelial cells, monocytes, mucosal epithelial cells, and stromal fibroblasts, can potentially harbor the virus. However, the detection of DCH material (e.g. by IHC) and actual infection of various tissues cannot be definitively claimed from these results, although they do suggest the possibility of a DCH replication site or a reservoir within these tissues. Further descriptions of DCH should now focus on the potential link to DCH replication associated with lesions to provide more information regarding any pathological lesions caused by this infection.

Within DCH associated with the diseased cats, the ultrastructure of DCH localization would not only provide a non-biased identification but also shed light on the intracellular

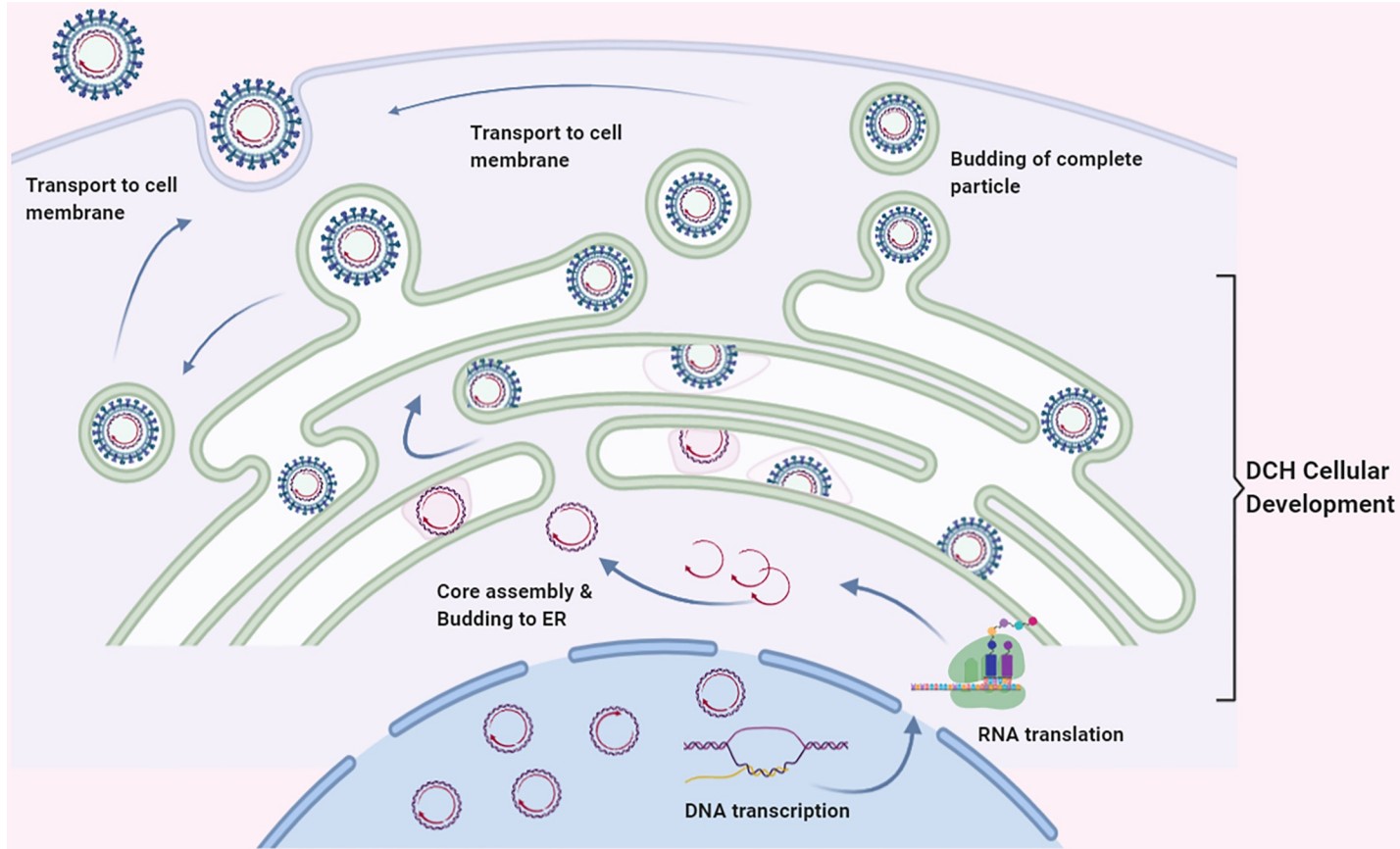

**Fig 8. Schematic representation of the cellular development of DCH.** The viral core particles are in the nucleus. After transcription/translation process, the DCH DNA is assembled in the cytoplasm and then buds to the ER. The process of DCH development occurs in the ER with invagination of the membrane to form tadpole-like phenomenon. The viral envelope is then developed with the activity of the ER membrane and finally transported to the plasma membrane to excrete the complete DCH virion as budding process.

activity of the virus [57]. Here, we described the intracellular localization and maturation of DCH particles in both liver and lung sections using TEM analysis. Similar to in the intracellular development of HBV, the presence of Dane particles in the tissue represents the complete development of the virion [58, 59]. We detected the DCH particles in the nucleus and/or cytoplasm of hepatocytes, which may suggest the initial localization of the virus as described in HBV [58, 60]. Surprisingly, we found viral particle protrusion from the nucleus into the cisternae of ER that was evidenced by coating the core particle with the ER membrane. Further evidences suggesting that the ER may be associated with DCH intracellular maturation was revealed by the attachment of incomplete Dane-like particles to the wall of the ER with stalks and the free floating of complete Dane-like particles in the cisternal cavity. Here, we schematically illustrate the possible DCH particle development in Fig 8, which is similar to that described in the maturation of HBV [58, 59] and other viruses, including rubella virus [61] and Zika virus [62].

Interestingly, DCH-immunopositive signals were also observed in extra-hepatic organs. This was further explored in lung sections by TEM analysis. The DCH particles were found in the bronchial gland epithelia and in fibroblast cells, which may suggest that these cells could support viral replication, as previously demonstrated in other hepadnaviral infection models [63, 64] and natural infection of the HBV associated with chronic hepatis [56]. However, this

interpretation for DCH needs further experimental investigation on the DCH biology and replication for verification.

In summary, this study identified DCH in cat sera, with the highest prevalence in FIV codetection. The nine Thai DCH strains sequenced here revealed genetic diversity within these DCH Thai strains that segregated into three novel monophyletic clades. In addition, evidence of recombination was found that may influence the evolution of DCH diversity. The hepatotropism and extra-hepatic manifestations of the virus were revealed with reference to intracellular development of DCH particles. Although the definitive role of pathogenicity of DCH remains undetermined, a contributory role of virus-associated systemic diseases is possible. Further elucidation on the influence of the virus on feline health is undoubtedly necessary.

## Supporting information

**S1 Raw image.**
(PDF)

**S1 Table. The DCH-specific primers used for the full-length genome characterization.** The primers were designed based on an alignment of the three DCH genomes that were available in GenBank.
(PDF)

**S1 File. Amplification protocols used for characterization of the full-length DCH genome.** The PCR amplifications were performed using the primers described in S1 Table.
(PDF)

**S2 File. Automated IHC protocol for DCH detection in FFPE tissues of DCH-PCR positive moribund cats.**
(PDF)

**S1 Fig. Negative controls for the DCH IHC staining.** No reaction is present within an incubation of normal rabbit IgG antibody NI01control in the (A) liver, (B) kidney, (C) lung, and (D) heart sections of cat no. 3, the (E) intestine of cat no. 2, and (F) brain of cat no. 1.
(TIF)

**S2 Fig. Domestic cat hepadnavirus particles in cytoplasm of the lung tissue.** Representative TEM image showing the ultrastructure of the bronchial epithelium and the diffuse electron-dense particles in the cytoplasm (inset). Bar indicates as in figure.
(TIF)

**S3 Fig. Domestic cat hepadnavirus particles in nucleus of the lung tissue.** Representative TEM image showing the ultrastructure of the bronchial glandular epithelium and the focal cluster of electron-dense particles in the nucleus (inset). Bar indicates as in figure.
(TIF)

## Acknowledgments

Outstanding western blotting techniques were supported by Tanida Tungchaisin and Assistant Professor Dr. Kittipong Rattanaporn, Faculty of Agro-Industry, Kasetsart University, Bangkok, Thailand.

## Author Contributions

**Conceptualization:** Chutchai Piewbang, Somporn Techangamsuwan.

**Data curation:** Chutchai Piewbang.

**Formal analysis:** Chutchai Piewbang, Tanit Kasantikul.

**Funding acquisition:** Suwimon Boonrungsiman, Somporn Techangamsuwan.

**Investigation:** Chutchai Piewbang, Sabrina Wahyu Wardhani, Surangkanang Chaiyasak, Jakarwan Yostawonkul, Poowadon Chai-in, Tanit Kasantikul.

**Methodology:** Chutchai Piewbang, Poowadon Chai-in, Suwimon Boonrungsiman.

**Project administration:** Chutchai Piewbang, Jakarwan Yostawonkul, Somporn Techangamsuwan.

**Resources:** Suwimon Boonrungsiman, Somporn Techangamsuwan.

**Software:** Chutchai Piewbang.

**Supervision:** Suwimon Boonrungsiman, Somporn Techangamsuwan.

**Validation:** Chutchai Piewbang, Somporn Techangamsuwan.

**Visualization:** Chutchai Piewbang, Tanit Kasantikul.

**Writing – original draft:** Chutchai Piewbang.

**Writing – review & editing:** Somporn Techangamsuwan.

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
