## [Decision Letter · Decision Letter 0]

24 Aug 2020

PONE-D-20-22914

Insights into the genetic diversity, recombination, and systemic infections with evidence of intracellular maturation of hepadnavirus in cats

PLOS ONE

Dear Dr. Piewbang,

Thank you for submitting your manuscript to PLOS ONE. After careful consideration, we feel that it has merit but does not fully meet PLOS ONE’s publication criteria as it currently stands. Therefore, we invite you to submit a revised version of the manuscript that addresses the points raised during the review process.

We look forward to receiving your revised manuscript.

Kind regards,

Haitao Guo

Academic Editor

PLOS ONE

Journal Requirements:

2.Thank you for stating the following in the Acknowledgments Section of your manuscript:

[C.P. was supported by the Ratchadapisek Somphot Fund for Postdoctoral Fellowship,

Chulalongkorn University. S.W. is financially afforded by scholarship program for ASEN

countries, Chulalongkorn University. This research was funded by The Thailand Research Fund

(RSA6180034), Grant for Joint Funding of External Research Project, Ratchadaphisek Somphot

Endowment Fund and Veterinary Science Research Fund (RES_61_364_31_037),

Chulalongkorn University, and Veterinary Pathogen Bank, Faculty of Veterinary Science,

Chulalongkorn University.]

 [The author(s) received no specific funding for this work.]

3. Your ethics statement must appear in the Methods section of your manuscript. If your ethics statement is written in any section besides the Methods, please move it to the Methods section and delete it from any other section. Please also ensure that your ethics statement is included in your manuscript, as the ethics section of your online submission will not be published alongside your manuscript.

Reviewers' comments:

Reviewer's Responses to Questions

**Comments to the Author**

1. Is the manuscript technically sound, and do the data support the conclusions?

Reviewer #1: Yes

Reviewer #2: Yes

2. Has the statistical analysis been performed appropriately and rigorously? 

Reviewer #1: N/A

Reviewer #2: Yes

3. Have the authors made all data underlying the findings in their manuscript fully available?

Reviewer #1: Yes

Reviewer #2: Yes

4. Is the manuscript presented in an intelligible fashion and written in standard English?

Reviewer #1: Yes

Reviewer #2: Yes

5. Review Comments to the Author

Reviewer #1: The manuscript described the detection of the recently discovered domestic cat hepadnavirus (DCH) in cat sera and cats died from hepatic diseases. The authors obtained and characterized the full-length viral genomes and revealed that nine DCH sequences phylogenetically formed three distinct monophyletic clades. They found a putative DCH recombinant strain. They also performed IHC staining of DCH and showed a wide-spread positivity of DCH in multiple tissues. The writing is clear and the results interesting.

1. Table 1, the viral copy number in various organs would be better presented as copies per milligram or gram of tissue.

2. A polyclonal anti-human HBc was used in the IHC staining. The authors should verify the cross-reactivity and specificity of this antibody to DCH core protein using Western blot.

Reviewer #2: The authors explored the genetic diversity of the recently discovered mammalian hepadnavirus, tentatively named domestic cat hepadnavirus (DCH), among a collection of cat sera and cat tissues using PCR. Their full-length genome characterization of revealed the nine Thai DCH sequences obtained formed three distinct monophyletic clades. They found a putative DCH recombinant strain, suggesting a possible role of recombination in DCH evolution. They also used qPCR to determine the viral genome copy numbers in various organs of the cats with liver diseases and were able to localize the putative viral capsid protein in tissues using a polyclonal antibody against the HBV capsid protein via immunohistochemistry (IHC). In addition to the liver, positive-DCH immunoreactivity was found in various other organs, including kidneys, lung, heart, intestine, brain, and lymph nodes, suggesting possible systemic infection.

This report adds some interesting new information regarding a relatively new mammalian hepadnavirus. I have a couple of questions for the authors.

1. Did the authors ever verify the specificity of the polyclonal antibody against the capsid by western blot analysis? Although the IHC control showed some evidence of specificity, western blot analysis will be more definitive. Inclusion of an alignment of the DCH and HBV capsid protein sequence would also help to evaluate the specificity of the polyclonal antibody.

2. The EM images are rather poor and not definitive. The putative capsids claimed by the authors might be surface antigen spheres, given the reported diameter of 23-25 nm, which is closer to the HBsAg spheres (ca. 22 nm) than the HBV capsids (ca. 30 nm). If feasible, immuno-EM using the cross-reactive capsid antibody would help to identify the capsid particles, which at the same time will also help to verify the specificity of the antibody.

6. PLOS authors have the option to publish the peer review history of their article (what does this mean?). If published, this will include your full peer review and any attached files.

Reviewer #1: No

Reviewer #2: No

---

## [Author Response · Author response to Decision Letter 0]

23 Sep 2020

Rebuttal response on a revision of PONE-D-20-22914

1. Is the manuscript technically sound, and do the data support the conclusions?

Reviewer #1: Yes

Reviewer #2: Yes

Response: Thank you for your positive reviews to our manuscript. 

2. Has the statistical analysis been performed appropriately and rigorously? 

Reviewer #1: N/A

Reviewer #2: Yes

Response: Thank you for your positive reviews to our manuscript. 

3. Have the authors made all data underlying the findings in their manuscript fully available?

Reviewer #1: Yes

Reviewer #2: Yes

Response: Thank you for your positive reviews to our manuscript. 

4. Is the manuscript presented in an intelligible fashion and written in standard English?

Reviewer #1: Yes

Reviewer #2: Yes

Response: Thank you for your positive reviews to our manuscript. 

5. Review Comments to the Author

Reviewer #1: The manuscript described the detection of the recently discovered domestic cat hepadnavirus (DCH) in cat sera and cats died from hepatic diseases. The authors obtained and characterized the full-length viral genomes and revealed that nine DCH sequences phylogenetically formed three distinct monophyletic clades. They found a putative DCH recombinant strain. They also performed IHC staining of DCH and showed a wide-spread positivity of DCH in multiple tissues. The writing is clear and the results interesting.

1. Table 1, the viral copy number in various organs would be better presented as copies per milligram or gram of tissue.

Response: Thank you for your positive reviews to our manuscript. We have revised the unit of the viral load to per gram of extracted tissue and re-calculated the viral copy number as a revision found in the Table 1. (Page 16, lines 364-365)

2. A polyclonal anti-human HBc was used in the IHC staining. The authors should verify the cross-reactivity and specificity of this antibody to DCH core protein using Western blot.

Response: Thank you for your concern and suggestion. We have tested the cross-reactivity of the anti-HBcAg antibody with the DCH-positive samples, compared to the result of HBV-infected tissue using western blot analysis. Furthermore, we have provided the specificity of the antibody by performing the western blotting with the extracted protein derived from DCH-negative liver samples (from cats with/without hepatitis). The details and results of the western blotting were given in the Methods (page 10, lines 221-241) and Results section (pages 19-20, lines 454-469), respectively. 

Reviewer #2: The authors explored the genetic diversity of the recently discovered mammalian hepadnavirus, tentatively named domestic cat hepadnavirus (DCH), among a collection of cat sera and cat tissues using PCR. Their full-length genome characterization of revealed the nine Thai DCH sequences obtained formed three distinct monophyletic clades. They found a putative DCH recombinant strain, suggesting a possible role of recombination in DCH evolution. They also used qPCR to determine the viral genome copy numbers in various organs of the cats with liver diseases and were able to localize the putative viral capsid protein in tissues using a polyclonal antibody against the HBV capsid protein via immunohistochemistry (IHC). In addition to the liver, positive-DCH immunoreactivity was found in various other organs, including kidneys, lung, heart, intestine, brain, and lymph nodes, suggesting possible systemic infection.

This report adds some interesting new information regarding a relatively new mammalian hepadnavirus. I have a couple of questions for the authors.

1. Did the authors ever verify the specificity of the polyclonal antibody against the capsid by western blot analysis? Although the IHC control showed some evidence of specificity, western blot analysis will be more definitive. Inclusion of an alignment of the DCH and HBV capsid protein sequence would also help to evaluate the specificity of the polyclonal antibody.

Response: Thank you for your positive review on our manuscript. As your suggestion, we have further elucidated the cross-reactivity throughout the specificity of the antibody used for IHC analysis by using the western blotting. Details and results of the western blotting were described in the Methods (page 10, lines 221-241) and Results section (pages 19-20, lines 454-469), respectively. 

2. The EM images are rather poor and not definitive. The putative capsids claimed by the authors might be surface antigen spheres, given the reported diameter of 23-25 nm, which is closer to the HBsAg spheres (ca. 22 nm) than the HBV capsids (ca. 30 nm). If feasible, immuno-EM using the cross-reactive capsid antibody would help to identify the capsid particles, which at the same time will also help to verify the specificity of the antibody.

Response: We sincerely respected with this comment since the EM images are quite not definitive. With the respects of TEM performing on the dead animals and fixative agents are not definitively appropriated for TEM analysis, resulting in poor-defined images. However, we performed the TEM from the FFPE section by locating the areas that were positive to the IHC analysis using Pop-off technique, supporting the particles found in those areas would be the targeting virus. With the respect of reviewer’s concern, we have revised that the detected particles would be the surface antigen spheres as suggestion by reviewer and also added further image of the EM. We honestly said that we are not able to do the immune-EM in this recent and we have performed the western blotting to test the specificity of the antibody used for IHC as described in Methods and Results sections. 

6. PLOS authors have the option to publish the peer review history of their article (what does this mean?). If published, this will include your full peer review and any attached files.

Do you want your identity to be public for this peer review? For information about this choice, including consent withdrawal, please see our Privacy Policy.

Reviewer #1: No

Reviewer #2: No

Response: We respected with the reviewer’s decision.

---

## [Editor Report · Decision Letter 1]

12 Oct 2020

Insights into the genetic diversity, recombination, and systemic infections with evidence of intracellular maturation of hepadnavirus in cats

PONE-D-20-22914R1

Dear Dr. Piewbang,

We’re pleased to inform you that your manuscript has been judged scientifically suitable for publication and will be formally accepted for publication once it meets all outstanding technical requirements.

Kind regards,

Haitao Guo

Academic Editor

PLOS ONE
---

## [Editor Report · Acceptance letter]

14 Oct 2020

PONE-D-20-22914R1 

Insights into the genetic diversity, recombination, and systemic infections with evidence of intracellular maturation of hepadnavirus in cats 

Dear Dr. Piewbang:

I'm pleased to inform you that your manuscript has been deemed suitable for publication in PLOS ONE. Congratulations! Your manuscript is now with our production department. 

Kind regards, 

on behalf of

Dr. Haitao Guo 

Academic Editor

PLOS ONE